# Development of the Follow-Up Human 3D Oral Cancer Model in Cancer Treatment

**DOI:** 10.3390/biotech12020035

**Published:** 2023-05-11

**Authors:** Kazuyo Igawa, Kenji Izumi, Yoshinori Sakurai

**Affiliations:** 1Neutron Therapy Research Center, Okayama University, Okayama 700-8558, Japan; 2Division of Biomimetics, Faculty of Dentistry & Graduate School of Medical and Dental Sciences, Niigata University, Niigata 951-8514, Japan; izumik@dent.niigata-u.ac.jp; 3Institute for Integrated Radiation and Nuclear Science, Kyoto University, Osaka 590-0494, Japan; sakurai.yoshinori.8n@kyoto-u.ac.jp

**Keywords:** 3D cancer model, preclinical study, cancer treatment, quality of life, multidisciplinary treatment

## Abstract

As function preservation cancer therapy, targeted radiation therapies have been developed for the quality of life of cancer patients. However, preclinical animal studies evaluating the safety and efficacy of targeted radiation therapy is challenging from the viewpoints of animal welfare and animal protection, as well as the management of animal in radiation-controlled areas under the regulations. We fabricated the human 3D oral cancer model that considers the time axis of the follow up in cancer treatment. Therefore, in this study, the 3D model with human oral cancer cells and normal oral fibroblasts was treated based on clinical protocol. After cancer treatment, the histological findings of the 3D oral cancer model indicated the clinical correlation between tumor response and surrounding normal tissue. This 3D model has potential as a tool for preclinical studies alternative to animal studies.

## 1. Introduction

Although ablative surgery is a major therapeutic modality, cancer treatment is performed in multidisciplinary modalities, combining local radiotherapy with systemic chemotherapy. Various radiation devices such as X-rays, proton beams, heavy particle beams, and neutron beams have been developed for radiation therapy for cancer [1]. Currently, targeted radiation therapy that combines the drugs within cancer tissue and reacts with radiation to induce cancer cell death at the cellular level has been developed in consideration of the quality of life (QOL) such as photoimmunotherapy [2] and Boron neutron capture therapy (BNCT) [3]. This targeted drug-radiation technology requires a comprehensive assessment of the drug, radiation, and the reaction of the drug and radiation as preclinical studies. Preclinical studies in humans are desirable, but there are ethical difficulties. Therefore, preclinical studies rely on animal models of human disease. However, preclinical animal studies determining the safety and efficacy of a new radiation therapy are challenging from the viewpoints of animal welfare and animal protection, as well as the management of animals in radiation-controlled areas [4].

The various three-dimensional (3D) models mimicking cancer microenvironments, such as spheroids and organoids, are promising for the suitable safety test compared with simple 2D models [5,6,7]. Therefore, various 3D cell culture systems for screening drugs have been developed, instead of 2D cell culture screening systems [8]. Regarding chemical safety and animal welfare, the skin corrosion test using a 3D human skin model has been committed to Organization for Economic Cooperation and Development (OECD) test guidelines [9]. Recently 3D tissue models for screening cancer radiotherapy have been developed [10,11].

In this study, we developed a 3D human oral cancer model focused on the time axis of the subsequent course of cancer treatment. Our 3D oral cancer model is composed of a stromal layer of type I collagen gel, approximately 3 mm in thickness, in which normal oral fibroblasts are repopulated, and an oral cancer layer is generated on top of it, which results in a vertical dimension in this model. This method recreates more physiologically relevant microenvironments for cells, and the collagen matrix (gel), as an extracellular matrix (ECM), provides structural support for constituting cells. This 3D model mimics the biochemical and cellular makeup of tumors in their microenvironments more closely than 2D cultures. Using our 3D oral cancer model consisting of human oral squamous carcinoma cells and normal oral mucosal fibroblasts, the safety and efficacy of cancer treatment were evaluated. As a cancer treatment, BNCT was chosen in this study because it requires comprehensive evaluation based on the boron agent as a chemical drug and neutron irradiation as radiotherapy.

## 2. Materials and Methods

### 2.1. Human Cell Culture

To develop the oral cancer model, human primary normal oral fibroblasts (NOFs; Niigata University) and the human oral squamous carcinoma cell lines (SAS; JCRB) were used. Both human cell lines were authenticated using short tandem repeat profiling in May 2020, and all experiments were performed with mycoplasma-free cells [12]. The cells were maintained in Dulbecco’s Modified Eagle Medium (DMEM; Thermo Fisher Scientific, Waltham, MA, USA) supplemented with 10% fetal bovine serum (FBS; Corning, New York, NY, USA), gentamicin (5.0 µg/mL), and amphotericin B (0.375 µg/mL; Thermo Fisher Scientific) under a humidified atmosphere of 5% CO_2_ at 37 °C.

### 2.2. Fabrication of Human 3D Oral Cancer Model

Collagen matrix solution (Nitta gelatin, Osaka, Japan) containing NOFs (1.67 × 10^5^ cells/mL) in a 6-well tissue culture insert (Greiner Bio, Roskilde, Denmark) was incubated with DMEM containing 10% FBS under a humidified atmosphere of 5% CO_2_ at 37 °C for 2 days. The composite of collagen matrices was then detached from the wall of the tissue culture insert using a 200-µL micropipette tip. After 5 days, SAS cells (5.0 × 10^5^ cells) were seeded on the top surface of the collagen matrix in which NOFs are embedded. The 3D cancer models were cultured under submerged conditions with DMEM containing 10% FBS for 7 days and then raised to an air–liquid interface from day 14 and cultured for an additional 7 days. The 3D oral cancer models were completed for 21 days in culture.

### 2.3. Treatment Planning of BNCT for 3D Oral Cancer Model

First, the boron concentration of 3D cancer model was measured. The cancer model administered with boron (Borofaran, Stella Pharma, Osaka, Japan) for 2 h was cleaned with PBS twice and heated after being treated with HNO_3_, measured by Inductively Coupled Plasma-Mass Spectrometer (ICP-MS, Agilent7500cx, Agilent Technologies, Tokyo, Japan) [13]. Subsequently, the neutron fluence at the tumor area and normal area in the 3D oral cancer model was measured by activation method using gold foil at Kyoto University Research Reactor (KUR) [14]. Based on the boron concentration and the neutron fluence, the optimal irradiation time for 3D cancer model was determined by clinical methods of BNCT. That is, it was calculated from the value of relative biological effectiveness (RBE, neutron: 3.0, gamma: 1.0) and compound biological effectiveness (CBE, cancer: 3.8, mucosa: 4.9) in BNCT [15].

### 2.4. BNCT for 3D Oral Cancer Model

Before neutron irradiation, boron agent (Borofaran, Stella Pharma, Japan) or saline (OTSUKA NORMAL SALINE, Tokyo, Japan), as placebo was added to the oral cancer model, respectively, and incubated for 2 h. Then, oral cancer models were irradiated by neutron at KUR and cultured for another 7 days.

### 2.5. Histological Analysis

After 7 days of BNCT, the 3D models were fixed with 10% formalin, embedded in paraffin, cut in 5 µm sections, and stained with hematoxylin and eosin staining. All images of the sections were obtained with an all-in-one fluorescence microscope (BZ-X800, Keyence, Osaka, Japan).

### 2.6. Statistical Analysis

The data are shown as means ± standard deviation (S.D.). The statistical differences were compared by one-way analysis of variance or Fisher’s exact test. A *p*-value of less than 0.05 was considered statistically significant.

## 3. Results

### 3.1. Fabrication of Human 3D Oral Cancer Model

The diagram of a hybrid human 3D oral cancer model consisting of human SAS cells and NOFs was shown (Figure 1, left panel). The histological examination showed the vertical proliferation of SAS cells over time (Figure 1, right panel). Just 1 day after the seeding of SAS cells, the cancer cells formed a monolayer at day 8 from the start of 3D model fabrication, and they generated a few layers at day 14 in a submerged condition. Finally, they developed a multilayer during an air-liquid interface for the last 7 days within 21 days in culture. The thickness of the stromal layer of the 3D oral cancer model, in which NOFs were proliferated, was 20.7 ± 9.3 mm on day 7, 8.9 ± 0.5 mm on day 14, and 3.7 ± 0.2 mm on day 21.

### 3.2. Treatment Planning of BNCT for 3D Oral Cancer Model

The boron concentration and the neutron fluence of the oral cancer model were measured before neutron irradiation. The boron concentrations of cancer cells and the stromal layer were 26.2 ± 0.3ppm and 5.9 ± 0.8 ppm, respectively, and the neutron fluence at cancer cells and stromal layer were 1.52 × 10^12^ cm^−2^ and 1.16 × 10^12^ cm^−2^ (Table 1). The irradiation time was determined to be 20 min, based on the tumor dose (>10 Gy-eq) [16] and the normal dose (<5Gy-eq) [17].

### 3.3. Evaluation of BNCT for 3D Oral Cancer Model

According to the treatment planning, the boron-administered 3D oral cancer models irradiated by a neutron for 20 min were cultured for another 7 days. The histological examinations revealed that the thickness of the cancer cells layer in the model that received BNCT reduced from the multilayer to a monolayer, although it stayed in the multilayer in the placebo group. In contrast, there was not significant difference in the thickness of stromal layers (Figure 2).

## 4. Discussion

Recently, a 3D model is known as a promising tool mimicking the tumor and its microenvironment, and it is applied for the drug and radiation screening system [6]. In this study, the 3D oral cancer model with normal oral fibroblasts and oral cancer cells was fabricated as the follow-up 3D oral cancer model. In our 3D oral cancer model, the overlying cancer cells proliferated on the stromal layers, resulting in an increase in the thickness of the oral cancer cells layer over time as opposed to a decrease in the thickness of the stromal layer. Our preliminary data indicated that the subsequent conditions of the treatment, including not only the tumor response but also the surrounding normal tissue, were obtained via this 3D oral cancer model before registration of the cancer treatment.

Oral cancer management is shifted from radical therapy to function-preserving therapy for QOL because the maintenance of oral functions such as mastication, swallowing, respiration, and vocalization is essential for a healthy life [18]. The targeted drug radiation therapy at the cellular level, in which chemical agents accumulated in cancer cells responded by radiation, is expected to be the next generation of minimally invasive cancer therapy because it has scarce adverse effects on surrounding normal cells and preserves oral functions. As one of the targeted drug radiation therapies, BNCT is applied for head and neck cancer because of highly selective tumor cell death induction without significant damage to normal tissue [3,15]. The principle of BNCT is that boron at a tumor cell is irradiated with a neutron, the producing energy affects a single cell area, and subsequently, a tumor cell was damaged at the cellular level with little damage to surrounding normal cells. BNCT is a promising treatment for oral cancer, helping cancer patients early return to social life. More than thirty BNCT projects have been initiated worldwide, but in 2020, accelerator-based BNCT for locally unresectable recurrent or unresectable advanced head and neck cancer was approved only in Japan [19]. It took a long time to obtain an approval for human clinical trial due to the complicated procedures, including the three approvals of the medical boron drug, neutron device, and treatment program. That is, the animal tests for the medical boron drug and neutron device were required as preclinical studies. The large number, the long breeding period, and the high cost of animals in developing BNCT will violate the principle of 3Rs (Replacement, Reduction, and Refinement) [20]. In addition, it is difficult to conduct animal management in a radiation-controlled area due to regulations, although the neutron has no charge, and its area of influence can be estimated before the treatment to reduce the damage to surrounding tissues for QOL. Therefore, the alternative animal model is urgently needed for BNCT. For verification of the model as a preclinical study, our 3D oral cancer model was received by BNCT in the same way as human oral cancer treatment and follow up. In this oral cancer model, boron concentration was measured by ICP-MS, and neuron fluence was measured by gold foil in the same preparation of BNCT cancer treatment in humans [21]. An indication of BNCT was the boron concentration in our 3D oral cancer model, which was over 25 ppm. Based on the treatment planning for the cancer model [22], we were able to observe the 3D cancer model for 2 weeks after BNCT. The histological analysis showed that the oral cancer cells were significantly damaged, and the proliferation of cancer cells was controlled by BNCT. Our findings showed that the invasion of cancer cells to surrounding tissues was suppressed, then the stromal layer was not damaged. Since the 3D oral cancer model could be treated as a malignant lesion in vivo by BNCT, this 3D oral cancer model has the potential to be indicative of the therapeutic efficacy of BNCT for the international standardization of BNCT [23].

Oral mucositis after BNCT frequently occurs in cases of recurrent cancer after chemoradiation therapy [18]. One of the reasons is based on the optimistic evaluation of intestinal mucosa instead of oral mucosa in animal studies because the causes of mortality and pain are not clear from nutritional disorder or overdose of treatment. Recently, the histopathological evaluation of oral mucositis in mice treated by conventional X-ray was reported [24]. There are several reports for 3D oral mucosa models to meet the quality control criteria of OECD guidelines valid for the skin model to prevent COVID-19 [25,26]. By the combination of this 3D oral cancer model and the existing oral epithelial model, both treatment effects and the adverse effects of cancer therapy could be measured directly. In further study, the development of histological criteria is necessary to evaluate toxicity to oral mucosa using a 3D oral cancer model. Additionally, there are several animal models and the 3D organoids of oral mucositis for the development of therapeutic drugs [27,28]. Comparing these 3D oral mucositis models, the causes of oral mucositis will be clearly determined by using our 3D oral cancer model after the treatment of chemotherapy or radiotherapy. Moreover, since there are individual differences in therapeutic effects and side effects even under the same treatment protocol, the therapeutic efficacy for each patient by fabricating the 3D model using cancer and normal cells derived from the patients will provide the optimal multidisciplinary treatment as personalized medicine.

## 5. Conclusions

In this study, the 3D oral cancer model that considers the time axis of the treatment procedure was fabricated with human oral cancer cells and normal oral fibroblasts. The efficacy of a 3D oral cancer model, instead of in vitro 2D study and in vivo animal study, was verified in BNCT as drug-radiation therapy. In this preliminary data, the 3D human oral cancer model would be a useful tool for preclinical studies in the development of drug-radiation cancer therapy. However, further studies are needed to evaluate the long-term effects of cancer treatment. In the near future, the 3D oral cancer model using oral cancer and normal mucosal cells derived from the patients would provide personalized medicine.

## Figures and Tables

**Figure 1 biotech-12-00035-f001:**
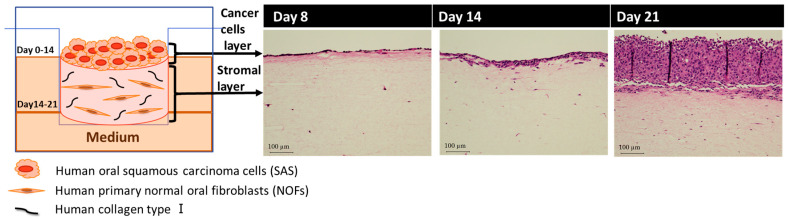
Fabrication of 3D oral cancer model. The proliferation of cancer cells was shown over time. The follow-up date is shown from the start of 3D model fabrication.

**Figure 2 biotech-12-00035-f002:**
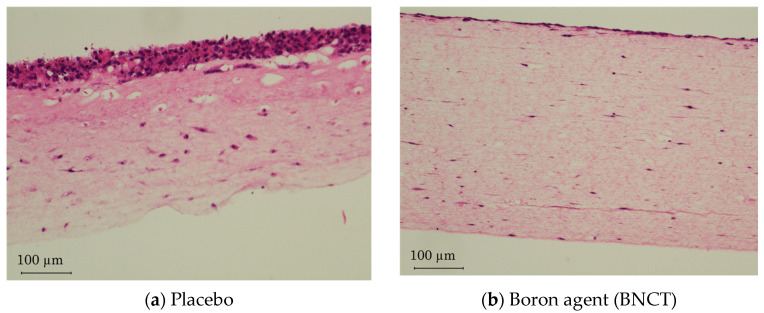
Evaluation for 3D oral cancer model at post-BNCT day 7. Before neutron irradiation, (**a**) placebo (saline) and (**b**) boron agent was administered. The histological examinations of the 3D oral cancer model at post-BNCT day 7 revealed that the thickness of cancer cells layer was significantly thinner compared with placebo.

**Table 1 biotech-12-00035-t001:** BNCT treatment planning.

Oral Cancer Model	Boron Concentration [ppm]	Thermal Neutron Fluence [cm^−2^]	* Total Dose [Gy-Eq]
Cancer cell layer	26.2 ± 0.3	1.52× 10^12^	13
Stromal cell layer	5.9 ± 0.8	1.16× 10^12^	3.7

* The total dose was calculated from relative biological effectiveness v (RBE) values (neutron: 3.0, gamma: 1.0) and CBE (cancer: 3.8, mucosa: 4.9).

## Data Availability

The data presented in this study are available from the corresponding author on reasonable requests.

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
