# Peer review of "Development of the Follow-Up Human 3D Oral Cancer Model in Cancer Treatment"

_biotech, 2023, doi:10.3390/biotech12020035_

Round 1

Reviewer 1 Report

Absract  should  be  more  concise  and  clear

Abstract conclusion should be in focus to the malignant mesothelioma

Elaborate  more  on malignant mesothelioma clinics, therapy

Figure 1 only  significant  figure  is  needed

Figure  2  only  one figure  is needed

Author Response

Dear Review and Editor

Thank you very much for reviewing our manuscript and offering valuable advice.

We have addressed your comments with point-by-point responses, and revised the manuscript accordingly.

Responses to the Comments by the Associate editor:

  1. Absract  should  be  more  concise  and  clear

Thank you for the advice. It is corrected as more concise and clear.

  1. Abstract conclusion should be in focus to the malignant mesothelioma

This study is based on the oral cancer, therefore we will not mention about the malignant mesothelioma. In the future, we will challenge for it.

  1. Elaborate  more  on malignant mesothelioma clinics, therapy

This study is based on the oral cancer. Therefore, we mention oral mucositis in discussion part.

4.Figure 1 only  significant  figure  is  needed

In Figure 1, the histological examination of this follow-up 3D oral cancer model was shown.

5.Figure  2  only  one figure  is needed

It is based on placebo-controlled comparative study using 3D oral cancer model. Then it is not enough for one figure.

Reviewer 2 Report

The manuscript entitled "Development of the follow-up human 3D oral cancer model in cancer treatment" submitted by Igawa et al. describes the fabrication of a 3D oral cancer model which would replaces the animal studies in preclinical studies. Please see the comment s below:

1) In the abstract, please expand QOL when its first appear in the text. also what a 3D model is. If it is a three-dimensional model, define how the model makes it three dimensional.

2) In line 46, please expand OECD and it's importance.

3) In figure 1, how does the medium transfers to stromal layer to grow the cells?  As the stromal cells lies at the bottom of the plate, is there a similar growth compared to cancer cells over 1-14 days? How the growth of stromal cells being measured over time, also is there a reason human collagen to be added to stromal layer?

4) In fig 1, Is there any layer between stromal and cancer layers? Please describe the 3D model very well, perhaps with the 3d figure.

5) In section 3.3, figure 2 needs to be revised in such a way that shows saline alone as control, stromal layer alone, cancer layer alone, stromal and cancer layer as combined before neutron irradiation and after 7 days post irradiation to observe the real difference.

6) Does BNCT affects any stromal cell damage in your model?

7) There are some grammatical errors in line 179 and 185, even in the rest of manuscript, please address them.

8) It would be ideal to conduct the study in animals (in vivo) and compare the results after BNCT in animals (if the oral cancer models exists) versus 3D oral cancer model which will validate the 3D model results.

9) In conclusions, please mention the advantages of 3D model over existing 2D model. Also point out any disadvantages.

Please address the revisions as major.

Author Response

Dear Editors and Reviewers

Thank you very much for reviewing our manuscript and offering valuable advice. We have addressed your comments with point-by-point responses, and revised the manuscript accordingly.

Responses to the Comments by the Associate editor:

  • In the abstract, please expand QOL when its first appear in the text.

also what a 3D model is.

Thank you for your advice.  The Quality of life (QOL) was corrected in Abstract and in line 33.

From 2020, the 3D is used in title and abstract as not only for the 3D model but also 3D culture, 3D printing, 3D images, 3D reconstruction et al. For example, 3D is used in abstract (Drost, J., Clevers, H. Organoids in cancer research. Nat Rev Cancer 18, 407–418 (2018). https://doi.org/10.1038/s41568-018-0007-6).  Then the abbreviation of 3D was indicated in line 41.

  • If it is a three-dimensional model, define how the model makes it three dimensional.

We will inset below in introduction ( Line 51).

Our 3D oral cancer model is composed of a stromal layer of type I collagen gel, approximately 3mm in thickness, in which normal oral fibroblasts are repopulated and an oral cancer cell layer that is generated on top of it, which results in a vertical dimension in this model. This method recreates more physiologically relevant microenvironments for cells. The collagen matrix (gel), as an extracellular matrix (ECM), provides structural support for constituting cells. This 3D model mimics the biochemical and cellular makeup of tumors in their microenvironments more closely than do 2D cultures.

  • In line 46, please expand OECD and it's importance.

Thank you for the advice. The OECD was corrected as Organization for Economic Cooperation and Development (OECD).

3) In figure 1, how does the medium transfers to stromal layer to grow the cells? 

Usually, the medium is passively transferred ,depending on cellular metabolic activity. The line of Medium is changed before Day 7 in figure 1.

  • As the stromal cells lies at the bottom of the plate, is there a similar growth compared to cancer cells over 1-14 days?

In the insert, NOFs are evenly distributed in the gel, as shown in the schematic and the photos om figure 1.

  • How the growth of stromal cells being measured over time,

The proliferation of NOFs is analyzed by histological analysis as shown figure 1.

  • also is there a reason human collagen to be added to stromal layer?

This is because normal oral submucosal tissue around oral cancer is composed of connective tissue, that is, mostly type I collagen.

  • In fig 1, Is there any layer between stromal and cancer layers?

NO.

  • Please describe the 3D model very well, perhaps with the 3d figure.

We believe that a cross-sectional image is sufficient to understand it.

9) In section 3.3, figure 2 needs to be revised in such a way that shows saline alone as control, stromal layer alone, cancer layer alone, stromal and cancer layer as combined before neutron irradiation and after 7 days post irradiation to observe the real difference.

Thank you for the great advice. We agree with you, however it is still in the preliminary stage and we have not tested it yet. We believe the further studies are needed.

 10)Does BNCT affects any stromal cell damage in your model?

We cannot give you a definite answer, because we believe that the analysis other than HE staining such as immunostaining and cellular metabolism is necessary.

11)There are some grammatical errors in line 179 and 185, even in the rest of manuscript, please address them.

Thank you for mention about typo and grammatical errors.

The previous article in line 179; In recently, the histopathologic grading of oral mucositis in mice treated by conventional X-ray was reported [24].

It is changed in line189 ; In recently, the histopathological evaluation of oral mucositis in mice treated by conventional X-ray was reported [24].

The previous articles in line 185;  In further study, it requires to histological criteria for scoring mucosal toxicity with 3D oral cancer model.

It is changed in line 195; In further study, histological criteria are necessary for the evaluation of mucosal toxicity using 3D oral cancer model.

12) It would be ideal to conduct the study in animals (in vivo) and compare the results after BNCT in animals (if the oral cancer models exists) versus 3D oral cancer model which will validate the 3D model results.

We totally agree with your comments.  In general, the animal model will be irradiation of ectopically implanted oral cancer cells, so it is more significant to use both together than to use the animal model for validation of the vitro model. We have also added that comments in discussion part.

13) In conclusions, please mention the advantages of 3D model over existing 2D model.

The advantages are as described in (1).

  • The ability to focus purely on the cross-talk between cancer cells and submucosal tissue.
  • The three-dimensional positioning of cells is similar to that of living organisms. 

14) Also point out any disadvantages.

The disadvantages are described in the discussion.

  • Long-term culture is difficult.
  • The relationship between cells may be too simple because there are no cells of immune and vascular system.

Reviewer 3 Report

To perform a human 3D oral cancer model for potential preclinical therapies with high animal ethics concern, the study successfully described a new approaching for cancer research. It is recommended that this manuscript should be published on BioTech.

However, there are some points that should be clarified.

1.    For historical analysis, the authors mentioned that sample was fixed and embedded and “cut in 5 mm sections”. Usually, historical samples were cut in 4-5μm size for the staining purpose. Is 5 mm too thick for H&E staining and imaging?

2.    The authors mentioned that: “The 3D oral cancer models were completed after culturing for 21 days.”. Why data only show 14 days? Is there any method for the tumor proliferation rate?

3.    There should be invasive and co-culture study for cancer therapy since chemicals and radiation may harm the adjacent cells as well as tissues.

Author Response

Dear Editors and Reviewers

Thank you very much for reviewing our manuscript and offering valuable advice. We have addressed your comments with point-by-point responses, and revised the manuscript accordingly.

Responses to the Comments by the Associate editor:

  1. For historical analysis, the authors mentioned that sample was fixed and embedded and “cut in 5 mm sections”. Usually, historical samples were cut in 4-5μm size for the staining purpose. Is 5 mm too thick for H&E staining and imaging?

Thank you for mention about typo. It is 5 micrometers, of course. We rechecked in PDF version.

  1. The authors mentioned that: “The 3D oral cancer models were completed after culturing for 21 days.”. Why data only show 14 days?

The follow-up date was shown from the start of 3D model fabrication in Figure1. Also, we  in materials and Methods ( Line 79 ) and in Result ( Line 112 ) as below.

Figure 1; The follow-up date is shown from the start of 3D model fabrication.

Line 79 ; The 3D cancer models were cultured under submerged conditions with DMEM containing 10% FBS for 7 days and then raised to an air–liquid interface from day 14 and cultured for an additional 7days. The 3D oral cancer models were completed after culturing for 21 days.

Line 112 ; Just 1 day after seeding of SAS cells, the cancer cells formed a monolayer at day 8 from the start of 3D model fabrication, and they generated a few layers at day 14 in a sub-merged condition. Finally, they developed a multilayer at day 21 during an air-liquid interface for 7 days.

  1. Is there any method for the tumor proliferation rate?

We believe that the analysis other than HE staining such as immunostaining and cellular metabolic activity or cell toxicity test is potential in this 3D oral cancer model.

  1. There should be invasive and co-culture study for cancer therapy since chemicals and radiation may harm the adjacent cells as well as tissues.

Thank you for the comment. It is still in the preliminary stage and the further studies using the 3 D oral cancer model including cancer-associated fibroblasts or a hybrid oral cancer model in which normal oral mucosa layer is surrounded by oral cancer layer are required for the validation of chemoradiotherapy.

Round 2

Reviewer 2 Report

  • In the abstract, please expand QOL when its first appear in the text.

also what a 3D model is.

Thank you for your advice. The Quality of life (QOL) was corrected in Abstract and in line 33.

From 2020, the 3D is used in title and abstract as not only for the 3D model but also 3D culture, 3D printing, 3D images, 3D reconstruction et al. For example, 3D is used in abstract (Drost, J., Clevers, H. Organoids in cancer research. Nat Rev Cancer 18, 407–418 (2018). https://doi.org/10.1038/s41568-018-0007-6). Then the abbreviation of 3D was indicated in line 41.

  • If it is a three-dimensional model, define how the model makes it three dimensional.

We will inset below in introduction (Line 51).

Our 3D oral cancer model is composed of a stromal layer of type I collagen gel, approximately 3mm in thickness, in which normal oral fibroblasts are repopulated and an oral cancer cell layer that is generated on top of it, which results in a vertical dimension in this model. This method recreates more physiologically relevant microenvironments for cells. The collagen matrix (gel), as an extracellular matrix (ECM), provides structural support for constituting cells. This 3D model mimics the biochemical and cellular makeup of tumors in their microenvironments more closely than do 2D cultures.

Reviewer: The model described here does not addresses the mixing of the stromal cell layer and oral cancer cell layer due to aggressiveness of cancer cells to grow faster. Fabricated model does not mimic the in vivo systems. Moreover, the authors need to verify the experimental findings in different settings.

  • In line 46, please expand OECD and it's importance.

Thank you for the advice. The OECD was corrected as Organization for Economic Cooperation and Development (OECD).

3) In figure 1, how does the medium transfers to stromal layer to grow the cells?

Usually, the medium is passively transferred,depending on cellular metabolic activity. The line of Medium is changed before Day 7 in figure 1.

  • As the stromal cells lies at the bottom of the plate, is there a similar growth compared to cancer cells over 1-14 days?

In the insert, NOFs are evenly distributed in the gel, as shown in the schematic and the photos om figure 1.

  • How the growth of stromal cells being measured over time,

The proliferation of NOFs is analyzed by histological analysis as shown figure 1.

  • also is there a reason human collagen to be added to stromal layer?

This is because normal oral submucosal tissue around oral cancer is composed of connective tissue, that is, mostly type I collagen.

  • In fig 1, Is there any layer between stromal and cancer layers?

NO.

  • Please describe the 3D model very well, perhaps with the 3d figure.

We believe that a cross-sectional image is sufficient to understand it.

9) In section 3.3, figure 2 needs to be revised in such a way that shows saline alone as control, stromal layer alone, cancer layer alone, stromal and cancer layer as combined before neutron irradiation and after 7 days post irradiation to observe the real difference.

Thank you for the great advice. We agree with you, however it is still in the preliminary stage and we have not tested it yet. We believe the further studies are needed.

Reviewer: The authors need to conduct and verify the experimental findings in defferent settings as suggested in order to validate the results. Figure 2 needs to be revised in such a way that shows saline alone as control, stromal layer alone, cancer layer alone, stromal and cancer layer as combined before neutron irradiation and after 7 days post irradiation to observe the real difference. Please conduct the experiments correctly and report the results as a new submission.

10Does BNCT affects any stromal cell damage in your model?

We cannot give you a definite answer, because we believe that the analysis other than HE staining such as immunostaining and cellular metabolism is necessary.

11There are some grammatical errors in line 179 and 185, even in the rest of manuscript, please address them.

Thank you for mention about typo and grammatical errors.

The previous article in line 179; In recently, the histopathologic grading of oral mucositis in mice treated by conventional X-ray was reported [24].

It is changed in line189 ; In recently, the histopathological evaluation of oral mucositis in mice treated by conventional X-ray was reported [24].

The previous articles in line 185; In further study, it requires to histological criteria for scoring mucosal toxicity with 3D oral cancer model.

It is changed in line 195; In further study, histological criteria are necessary for the evaluation of mucosal toxicity using 3D oral cancer model.

12) It would be ideal to conduct the study in animals (in vivo) and compare the results after BNCT in animals (if the oral cancer models exists) versus 3D oral cancer model which will validate the 3D model results.

We totally agree with your comments. In general, the animal model will be irradiation of ectopically implanted oral cancer cells, so it is more significant to use both together than to use the animal model for validation of the vitro model. We have also added that comments in discussion part.

13) In conclusions, please mention the advantages of 3D model over existing 2D model.

The advantages are as described in (1).

  • The ability to focus purely on the cross-talk between cancer cells and submucosal tissue.
  • The three-dimensional positioning of cells is similar to that of living organisms. 

14) Also point out any disadvantages.

The disadvantages are described in the discussion.

  • Long-term culture is difficult.
  • The relationship between cells may be too simple because there are no cells of immune and vascular system.

The experimental model fabrication does not address the major revisions  and especially figure 2 needs to be revised with experimental conditions suggested in order to be validated.

Author Response

Thank you for your ongoing consideration of our manuscript.

The model described here does not addresses the mixing of the stromal cell layer and oral cancer cell layer due to aggressiveness of cancer cells to grow faster. Fabricated model does not mimic the in vivo systems. Moreover, the authors need to verify the experimental findings in different settings.

We respectively disagree with you. According to our previous study (REF.12), phenotypic transformation from normal oral fibroblasts to CAFs (cancer associated fibroblasts) is a key factor for “invasiveness” of oral cancer. Therefore, we do NOT think that just mixing normal oral fibroblasts with oral cancer cells provides the aggressiveness of cancer cells (hopefully you mean “invasiveness”) In contrast, oral cancer tissue in vivo does not necessarily acquire “invasiveness” at an early stage, and maintain the histological structure of normal oral mucosa (an epithelial layer and the underlying stromal layer). Therefore, we believe the histological feature of our 3D oral cancer model is similar to that of oral cancer tissue in vivo at an early stage.

The authors need to conduct and verify the experimental findings in defferent settings as suggested in order to validate the results. Figure 2 needs to be revised in such a way that shows saline alone as control, stromal layer alone, cancer layer alone, stromal and cancer layer as combined before neutron irradiation and after 7 days post irradiation to observe the real difference. Please conduct the experiments correctly and report the results as a new submission.

Thank you for your valuable comments again, especially suggesting a control using saline alone. We completely agree with you. Although the comprehensive study is necessary as you suggested, this experiment is still in the preliminary stage. To accelerate developing a 3D in vitro model using human cells, instead of animal studies, we need to test the feasibility of our model first, as a tool for analyzing the modality of cancer treatment such as BNCT. As a result, it was successful. For the next step, it is necessary for us to validate our 3D model for research use. Therefore, currently, we do not include another data in this paper with the revision of Fig 2.

The experimental model fabrication does not address the major revisions  and especially figure 2 needs to be revised with experimental conditions suggested in order to be validated.

Thank you very much for your insightful suggestion again. Although the comprehensive study is necessary as you suggested, this experiment is still in the preliminary stage. In further study, we will get the chance for the rare neutron irradiation time to validate the various conditions in 3D model.

Reviewer 3 Report

The revised manuscript is okay.

Author Response

Thank you very much for your positive comments. The manuscript was checked by a native English-speaking colleague again.